# Study on the Vertical Stability of Drilling Wellbore under Optimized Constraints

Ruixue Pan [1] , Jimin Liu [1,2,*], Hua Cheng [1,2] and Haixu Fan [1]

1   School of Civil Engineering and Architecture, Anhui University of Science and Technology, Huainan 232001, China; 2021200383@aust.edu.cn (R.P.); hcheng@aust.edu.cn (H.C.); fhx1323086021@126.com (H.F.)
2   Engineering Research Center of Underground Mine Engineering, Ministry of Education, Anhui University of Science and Technology, Huainan 232001, China
*   Correspondence: jimliu@aust.edu.cn

**Abstract:** With the development of coal resource extraction and wellbore construction proceeding towards deeper depths, the stability of drilling wellbore structures has become increasingly severe, even posing a barrier to the use of drilling method technology in deep wellbore construction. To address this issue, this study raised an optimized constraints method involving pre-throwing cement slurry to the bottom before wellbore decent, altering bottom constraints. Firstly, the critical depth and instability criterion of this optimized method was derived by catastrophe theory. Subsequently, the role of single-factor and multi-factor sensitivity analyses on critical depth was discussed. The engineering effects of optimized constraint methods were contrasted and examined in several drilling projects. Finally, the characteristic values of real engineering were computed using numerical techniques and ABAQUS2020 software, and the efficacy of optimization approaches was examined and validated. The results revealed that the critical depth increased by 41.39 ± 5%. The influence factors described in order of the degree were the counterweight water height, the elastic modulus, the thickness of the wellbore, and the self-weight of the wellbore, sequentially. The conclusion on structural stability between the numerical calculation solution and theoretical calculation solution was completely the same. The optimized constraints method can effectively improve the stability of the wellbore structure.

**Keywords:** optimized constraints method; deep drilled wellbore; cusp catastrophe theory; vertical stability

## 1. Introduction

As one of the major fossil energy sources, coal has brought a huge supply to power, steel, chemical, and other industries with the growing global demand for energy [1,2]. Coal mine wellbore construction remains an important project at present. However, during the process of drilling method construction, after the drilling rig has drilled the well hole, the builders will measure the deviation of the well hole and then place the prefabricated bottom of the well wall and each section of the well wall into the well hole, then connect them with flanges in turn. The resulting well wall structure is a round tubular structure, which is referred to as the wellbore structure by many scholars [3–5]. The wellbore structure is prone to longitudinal instability during the construction process in drilling methods. Once the wellbore structure loses its stability, it can lead to collapse accidents, causing significant personnel and economic losses [6–10]. Moreover, the depth of coal resource extraction in western China is gradually increasing, and the construction depth of new shafts is about to reach a historic height [11,12]. Therefore, the stability of drilling wellbore structures is an urgent engineering problem to be solved.

The drilling method is an advanced directional drilling technology that is widely used in energy extraction, mineral exploration, and rescue [13–16]. In general, the drilling

method is divided into three stages: floating, floated to the bottom but not filled and fixed, and filled and fixed. The second stage is the weakest stage, and it is easy for the wellbore to experience instability once there is any minor interference. Many scholars have conducted numerous studies on the stability characteristics of this stage. Hong [17,18] first proposed the problem of structural instability in the wellbore and analyzed structural stability treating it as a slender rod with hinged ends. Niu et al. [19,20] established the critical depth of the equal cross-section wellbore structure on full water and non-full water by the principle of minimum potential energy and energy method. Cheng et al. [21] further established the critical depth of the variable cross-section drilling wellbore structure. Liu et al. [22,23] established the critical depth of the equal cross-section drilling wellbore structure using catastrophe theory considering the sudden and irreversible characteristics of the wellbore instability process. Rong et al. [24] used numerical calculation methods to analyze the vertical stability of the entire drilling process. Cheng et al. [25] monitored the lateral displacement during construction on site. In summary, a relatively large amount of research on drilling wellbore structural stability has been accumulated. However, these studies only focus on the structural stability of traditional construction methods without proposing effective optimization measures to improve structural instability and conducting targeted research on optimization measures.

In this study, an optimization measure will be raised, and a targeted theoretical and numerical study on optimization measures will be carried out. Considering that wellbore structure is most prone to instability during the moment of suspended sinking and touching the bottom, the contact conditions at the bottom of the wellbore will directly affect structural stability controlling. So, we raise the idea of pre-throwing cement slurry to the bottom before wellbore falls to the bottom. In this optimized method, cement mortar is injected into the bottom by grouting pipe before the wellbore sinks to the bottom. The injection amount and the speed of counterweight water control the speed of the wellbore sinking. When the wellbore sinks to the bottom under the action of counterweight water, the cement has already been pre-thrown. At this time, the bottom of wellbore structure is placed on the pre-casted cement and firmly restricts the displacement in all directions, forming a fixed contact at the bottom. This optimized construction measure improves the structural stability of wellbore structure at the moment when the suspension sinks to the bottom by changing the constraint state from hinged to fixed.

In order to further study and verify the rationality and effectiveness of the optimized constraints method, a series of theoretical and numerical analysis studies were conducted. In terms of theoretical research, the mechanical characteristics of the wellbore structure under optimization measures were analyzed and obtained the total potential energy function and catastrophic instability model. Based on catastrophe theory, the sufficient and necessary conditions for wellbore instability were analyzed and obtained the critical depth. Based on current drilling engineering, the changes in critical depth before and after optimization measures were compared. The influencing factors and degree of critical depth after using optimization measures were also analyzed. In terms of numerical research, a numerical calculation method for vertical stability characteristic values was established, and the finite element model was established. Optimization measures were calculated and implemented according to actual engineering. In order to make it easier for readers to quickly understand the technical terms and abbreviations used in the text and to refer to them easily, a list of symbols has been added here to explain what the symbols represent. For more details, please refer to Table 1.

**Table 1.** Glossary of symbols and their meanings.

| Symbols | Meanings | Symbols | Meanings |
|---------|----------|---------|----------|
| $y$ | The deflection curve function of wellbore | $\Pi$ | The total potential energy of wellbore |
| $y'$ | The first derivative of $y$ | $U$ | Elastic strain energy |
| $y''$ | The second derivative of $y$ | $V$ | Total external potential energy |
| $\delta$ | The maximum deformation value of the deflection curve | $E$ | Elastic modulus of the material used for wellbore |
| $D$ | The outer diameter of wellbore | $I$ | Sectional moment of inertia of wellbore |
| $d$ | The inner diameter of wellbore | $A_1$ | The coefficient of $\delta$ |
| $k$ | Radius of curvature | $A_2$ | The coefficient of $\delta^2$ |
| $M$ | The in-plane bending moment | $A_4$ | The coefficient of $\delta^4$ |
| $\rho$ | The ratio of reinforcement | $x$ | The state variable of the system |
| $q_c$ | Self-weight of wellbore | $m$ | A control variable of the system |
| $P_m$ | Lateral pressure of mud on the external surface of wellbore | $n$ | A control variable of the system |
| $P_w$ | Lateral pressure of counterweight water on the internal surface of wellbore | $\Pi'(x)$ | The first derivative of $\Pi$ |
| $R_A$ | Counterforce on the bottom of wellbore | $\Pi''(x)$ | The second derivative of $\Pi$ |
| $R_B$ | The counterforce of wellhead support | $W_c$ | The external potential energy change due to self-weight |
| $H$ | Depth of wellbore | $W_m$ | The change in external potential energy due to the lateral pressure of mud |
| $\gamma_c$ | The weight of concrete | $W_w$ | The change in external potential energy under the lateral pressure of counterweight water |
| $\gamma_s$ | The weight of steel | $(W_m)_x$ | The horizontal component of $W_m$ |
| $\gamma_m$ | The weight of mud | $(W_m)_y$ | The vertical component of $W_m$ |
| $q_m$ | Gravity of mud per unit length on the outside of wellbore. | $(W_w)_x$ | The horizontal component of $W_w$ |
| $\gamma_w$ | The weight of counterweight water | $(W_w)_y$ | The vertical component of $W_w$ |
| $q_w$ | Gravity of counterweight water per unit length on the inside of wellbore. | $\Delta$ | Stability criterion |
| $H_w$ | Height of counterweight water | $\triangle$ | Thickness of wellbore |

## 2. Theoretical Methodology

### 2.1. Force Analysis of Wellbore under Optimized Construction Method

Before the wellbore is suspended and sinks to the bottom, the grouting pipe is used to fill the cement slurry on the bedrock surface so that the bottom of wellbore can fall on the bedrock surface. The wellbore's external side is wrapped by cement slurry, and the bottom of the wellbore is in contact with the bedrock surface. It can be assumed that the wellbore is regarded as a slender pressure rod with an upper hinge support and a lower end fixed support. For the convenience of this discussion, the wellbore is regarded as the whole of the continuity of a homogeneous nature, and we made the following basic assumptions:

(1) The material of the wellbore is completely elastic, obeying the law of Huke;
(2) The displacement and deformation of the wellbore during flexion is small;
(3) The effects of friction and bonding between the wall and mud are not taken into account.

The drilling wellbore is regarded as a slender rod hinged at the upper end and solidly supported at the lower end, so the deflection curve function of the drilling wellbore $y$ can be expressed as [26]

$$y = \sum_{n=1}^{\infty} C_n \left( \cos \frac{2n+1}{2H} \pi x - \cos \frac{2n-1}{2H} \pi x \right) \tag{1}$$

where $n$ is the number of the half wave number of the flexural curve function; $C_n$ is a coefficient to be determined; $x$ is the height at any point on the wellbore; and $H$ is the

length of the wellbore. Based on the assumption (2), taking $n$ equal to 1, the deflection curve deformation function of the drilling wellbore $y$ can be obtained as

$$y = \delta \left( \cos \frac{3\pi x}{2H} - \cos \frac{\pi x}{2H} \right) \tag{2}$$

where $\delta$ is the maximum deformation value of the deflection curve.

When the wellbore sinks to the bottom, the forces on the wellbore structure include the counterforce of wellhead support $R_B$, the vertically downward self-weight of the wellbore $q_c$, the lateral pressure of mud on the external surface of the wellbore $P_m$, the lateral pressure of counterweight water on the internal surface of the wellbore $P_w$, and the counterforce on the bottom of the wellbore $R_A$. These forces are shown in Figure 1.

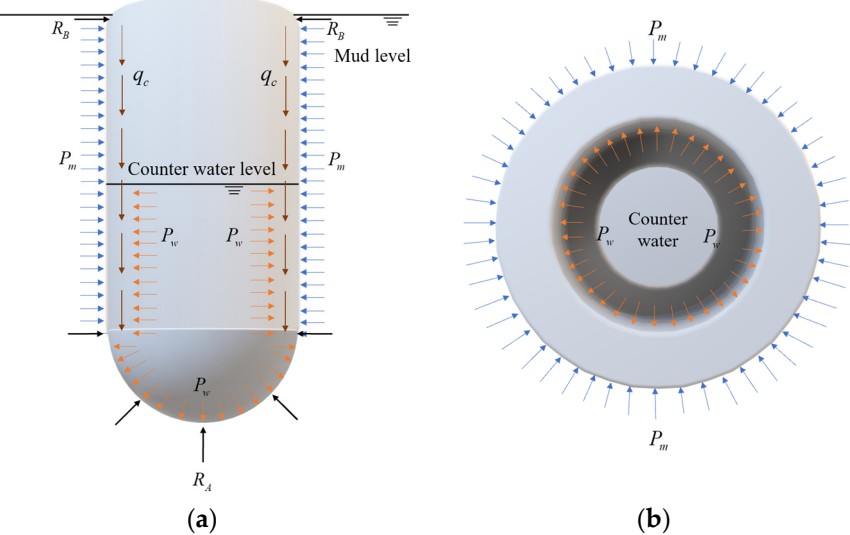

(**a**)            (**b**)

**Figure 1.** Force analysis model of drilling wellbore under non-full water conditions. (**a**) Longitudinal section force diagram. (**b**) Top view of force diagram.

The simplified force analysis of the wellbore is shown in Figure 2.

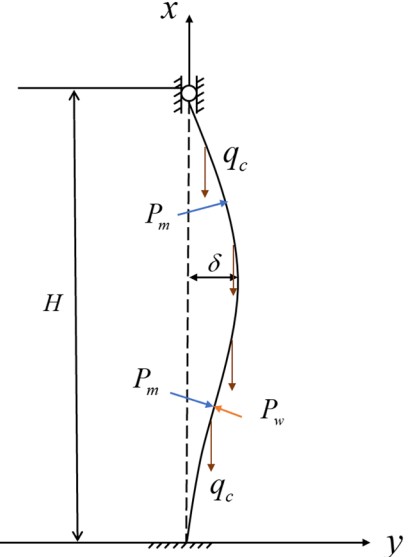

**Figure 2.** Simplified force analysis diagram of wellbore structure.

For the wellbore, its curvature equation is

$$\frac{1}{k} = \frac{y''}{\left[1 + (y')^2\right]^{\frac{3}{2}}} \tag{3}$$

where $y'$ is the first derivative of $y$ and $y''$ is the second derivative of $y$. $(y')^2$ is much less than 1, so it can be negligible, and the curvature equation is

$$\frac{1}{k} \approx y'' \tag{4}$$

Therefore, the in-plane bending moment $M$ is

$$M = \frac{1}{k} = \frac{y''}{\left[1 + (y')^2\right]^{\frac{3}{2}}} \approx y'' \tag{5}$$

From the assumption (2), it can be seen that the positive angle between the outer normal of any point on the shaft axis and the X-axis $\alpha$ is approximately equal to zero, and the bending deformation is small, so in the simplified calculation, it can be considered that

$$\sin \alpha \approx \alpha \approx \frac{\mathrm{d}y}{\mathrm{d}x}, \cos \alpha \approx 1, \alpha^2 \approx 0 \tag{6}$$

The wellbore weight per unit length $q_c$ is

$$q_c = \frac{\pi}{4} \left(D^2 - d^2\right) [\gamma_c (1 - \rho) + \gamma_s \rho] \tag{7}$$

where $D$ is the outer diameter of the wellbore; $d$ is the inner diameter of the wellbore; $\rho$ is the ratio of reinforcement; and $\gamma_c$ and $\gamma_s$ are the weight of the concrete and rebar, respectively. The direction of $q_c$ is vertically downward.

The lateral pressure of mud per unit length on the outside of the wellbore $P_m$ is

$$P_m = \frac{\pi}{4} D^2 \gamma_m \sin \alpha = q_m \sin \alpha \tag{8}$$

where $\gamma_m$ is the weight of mud and $q_m$ is the gravity of mud per unit length on the outside of the wellbore. The direction of $P_m$ is inward to the wellbore along the normal direction of a point.

The lateral pressure of counterweight water per unit length on the inside of the wellbore $P_w$ is

$$P_w = \frac{\pi}{4} d^2 \gamma_w \sin \alpha = q_w \sin \alpha \tag{9}$$

where $\gamma_w$ is the weight of counterweight water and $q_w$ is the gravity of counterweight water per unit length on the inside of the wellbore. The direction of $P_w$ is outward to the wellbore along the normal direction of a point.

### 2.2. Building the Total Potential Energy Function

From the force analysis of the wellbore in Section 2.1, the total potential energy equation of the drilled wellbore structure is obtained [27]

$$\Pi = U + V \tag{10}$$

where $U$ is the elastic strain energy released by the vertical flexural deformation of the structure and $V$ is the total external potential energy, which consists of the following components: $W_c$, the change in the external potential energy of the wellbore structure caused by the self-weight; $W_m$, the change in the external potential energy due to mud on

the outside of the wellbore; and $W_w$, the change in the external potential energy due to counterweight water on the inside of the wellbore. Since no displacement occurs under the function of forces $R_A$ and $R_B$, the potential energy generated by the function of forces $R_A$ and $R_B$ is zero. Therefore, the total potential energy equation for the wellbore structure $\Pi$ can be written as

$$\Pi = U + W_c + W_m + W_w \tag{11}$$

The elastic strain energy of wellbore cylinder is

$$U = \frac{1}{2}\int_0^H EI(y'')^2 \mathrm{d}x = \frac{41EI\pi^4}{32H^3}\delta^2 \tag{12}$$

where *E* represents the elastic modulus of the material used for the wellbore structure. Since the wellbore structure can be either fully reinforced concrete or steel plate concrete, the elastic modulus of the wellbore structure needs to be determined based on the structural form of the wellbore. For commonly encountered reinforced concrete wellbore structures, the elastic modulus of the material used for the wellbore structure *E* is the composite elastic modulus of the steel bar and concrete converted according to the reinforcement ratio; *I* is the sectional moment of inertia of the wellbore.

The external potential energy change due to self-weight is

$$W_c = -\frac{1}{2}\int_0^H q_c(H-x)(y')^2\mathrm{d}x = \left(1 - \frac{5}{16}\pi^2\right)q_c\delta^2 \tag{13}$$

The change in external potential energy due to the lateral pressure of mud is

$$W_m = (W_m)_x + (W_m)_y \tag{14}$$

$$(W_m)_x = \int_0^H (P_m)_x \cos\alpha \,\mathrm{d}x = \int_0^H q_m\sin\alpha\cos\alpha \,\mathrm{d}x = 0 \tag{15}$$

$$(W_m)_y = \int_0^H P_m\sin\alpha x\frac{1}{2}(y')^2\mathrm{d}x = \int_0^H q_m(\sin\alpha)^2x\frac{1}{2}(y')^2\mathrm{d}x = q_m\delta^4\pi^2\left(\frac{183\pi^2}{256H^2} + \frac{308}{75}\right) \tag{16}$$

The change in external potential energy under the lateral pressure of counterweight water is

$$W_w = (W_w)_x + (W_w)_y \tag{17}$$

$$(W_w)_x = -\int_0^{H_w} P_w\cos\alpha y\,\mathrm{d}x = -\int_0^{H_w} q_w\sin\alpha\cos\alpha y\,\mathrm{d}x = -2q_w\sin^2\frac{\pi H_w}{2H}\sin^2\frac{\pi H_w}{H}\delta^2 \tag{18}$$

$$(W_w)_y = -\int_0^{H_w} P_w\sin\alpha x\frac{1}{2}(y')^2\mathrm{d}x = -\int_0^{H_w} q_w(\sin\alpha)^2x\frac{1}{2}(y')^2\mathrm{d}x$$
$$= -\frac{\pi^4 q_w\delta^4}{32H^2}\left\{ \begin{array}{l} -\frac{1}{\pi^2}\left[\frac{17{,}657}{1200}\cos\frac{\pi H_w}{H} - \frac{9913}{600}\cos\frac{2\pi H_w}{H}\right.\\[4pt] \left.+9\left(-\frac{1267}{120} + \frac{549}{1200}\cos\frac{3\pi H_w}{H} + \frac{1}{100}\cos\frac{4\pi H_w}{H} + \frac{1}{8}\cos\frac{5\pi H_w}{H}\right)\right]\sin^2\frac{\pi H_w}{2H}\\[4pt] \frac{15}{\pi}\left(-\frac{59}{15}\sin\frac{\pi H_w}{H} + \frac{83}{48}\sin\frac{2\pi H_w}{H} - \frac{37}{30}\sin\frac{3\pi H_w}{H} + \frac{27}{80}\sin\frac{4\pi H_w}{H}\right.\\[4pt] \left.-\frac{9}{50}\sin\frac{5\pi H_w}{H} + \frac{9}{80}\sin\frac{6\pi H_w}{H}\right)\frac{H_w}{H} + \frac{183}{8}\left(\frac{H_w}{H}\right)^2 \end{array}\right\} \tag{19}$$

set

$$A = -\frac{1}{\pi^2}\left[\frac{17{,}657}{1200}\cos\frac{\pi H_w}{H} - \frac{9913}{600}\cos\frac{2\pi H_w}{H} + 9\left(-\frac{1267}{120} + \frac{549}{1200}\cos\frac{3\pi H_w}{H} + \frac{1}{100}\cos\frac{4\pi H_w}{H} + \frac{1}{8}\cos\frac{5\pi H_w}{H}\right)\right]\sin^2\frac{\pi H_w}{2H}$$
$$\frac{15}{\pi}\left(-\frac{59}{15}\sin\frac{\pi H_w}{H} + \frac{83}{48}\sin\frac{2\pi H_w}{H} - \frac{37}{30}\sin\frac{3\pi H_w}{H} + \frac{27}{80}\sin\frac{4\pi H_w}{H} - \frac{9}{50}\sin\frac{5\pi H_w}{H} + \frac{9}{80}\sin\frac{6\pi H_w}{H}\right)\frac{H_w}{H} + \frac{183}{8}\left(\frac{H_w}{H}\right)^2 \tag{20}$$

Then, Equation (19) can then be simplified as

$$(W_w)_{\mathrm{y}} = -\frac{A\delta^4\pi^4}{32H^2}q_w \tag{21}$$

According to Equations (12)–(21), the total potential energy function of the system can be obtained

$$\Pi = \left(-\frac{A\pi^4 q_w}{32H^2} + q_m\pi^2\left(\frac{183\pi^2}{256H^2} + \frac{308}{75}\right)\right)\delta^4 + \left\{\frac{41EI\pi^4}{32H^3} + \left(1 - \frac{5}{16}\pi^2\right)q_c - 2q_w\sin^2\frac{\pi H_w}{2H}\sin^2\frac{\pi H_w}{H}\right\}\delta^2 + 0\delta \tag{22}$$

An equivalent transformation of the Equation (22) yields

$$A_1 = 0 \tag{23}$$

$$A_2 = \frac{41EI\pi^4}{32H^3} + \left(1 - \frac{5}{16}\pi^2\right)q_c - 2q_w\sin^2\frac{\pi H_w}{2H}\sin^2\frac{\pi H_w}{H} \tag{24}$$

$$A_4 = -\frac{A\pi^4 q_w}{32H^2} + q_m\pi^2\left(\frac{183\pi^2}{256H^2} + \frac{308}{75}\right) \tag{25}$$

Then, set

$$x = 4\sqrt{4A_4}\cdot\delta \tag{26}$$

$$m = \frac{A_2}{\sqrt{A_4}} \tag{27}$$

$$n = \frac{A_1}{\sqrt[4]{4A_4}} \tag{28}$$

Then, the total potential energy function of the system can be written as

$$\Pi = \frac{1}{4}x^4 + \frac{1}{2}mx^2 + nx \tag{29}$$

which shows that the form of the potential function satisfies the standard form of the cusp catastrophe model [28,29]. In Equation (29), $x$, $m$, and $n$ are parameters in the standard form of the total potential energy function of the cusp catastrophe model, $x$ is the state variable of the system, and $m$ and $n$ are two control variables of the system. Set the first-order derivative of the total potential energy function equal to zero, then

$$\Pi'(x) = x^3 + mx + n = 0 \tag{30}$$

Those points whose first-order derivative of the total potential energy function is zero are singularities, also known as critical points. The surface formed by the set of critical points is called the equilibrium surface, as shown in Figure 3.

However, critical points do not necessarily guarantee that the system is stable. Only the point at which the total energy function takes a unique extreme value is stable, so the set of non-isolated singularities also requires that the second-order derivative of Equation (29) is also zero, so

$$\Pi''(x) = 3x^2 + m = 0 \tag{31}$$

From Equations (30) and (31), we can obtain the equation of bifurcation points set B

$$\Delta = 4m^3 + 27n^2 = 0 \tag{32}$$

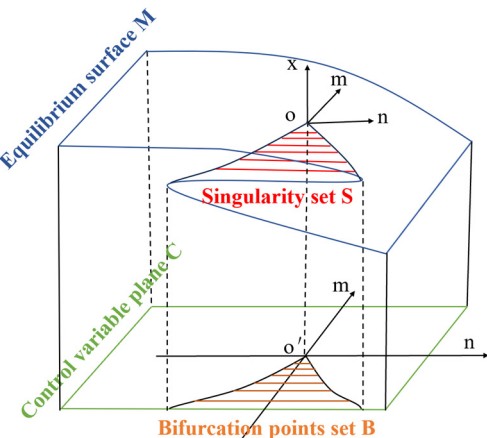

**Figure 3.** Equilibrium planes and control variable planes of the cusp catastrophe model.

*2.3. Derivation of Equations for Critical Depth*

2.3.1. Necessary Condition

It has been seen from Figure 3 that the structure is in an unstable state when $m \leq 0$, which means that the points appear inside the bifurcation points set B. For Equation (22), when $\Delta = 0$, it has two identical roots out of three roots, which is the critical state for the system to mutate. When $\Delta > 0$, it has only one real root and two complex roots, at which time we consider the system to be stable. When $\Delta < 0$, it has three real roots, at which time the system will exceed the critical state and then mutate. Using the discriminant $\Delta$, we can also determine whether the system is stable or not. From Equations (27), (28) and (32), it can be determined that the destabilization condition of the wellbore structure is

$$\Delta = 4m^3 + 27n^2 = 4\left(\frac{A_2}{\sqrt{A_4}}\right)^3 \leq 0 \tag{33}$$

from which it can be determined that

$$A_2 \leq 0 \tag{34}$$

By substituting Equation (24) into Equation (34), we can obtain

$$H \geq \sqrt[3]{\frac{41EI\pi^4}{(10\pi^2 - 32)q_c + 64q_w \sin^2 \frac{\pi H_w}{2H} \sin^2 \frac{\pi H_w}{H}}} \tag{35}$$

Equation (35) is a necessary condition for wellbore structure instability. The results show that vertical wellbore structure instability will occur as long as the depth of the wellbore is within its range. In addition, when Equation (35) obtains an equal sign, it is the critical condition for shaft wall structure instability, so the critical depth of shaft wall instability is

$$H_{cr} = \sqrt[3]{\frac{41EI\pi^4}{(10\pi^2 - 32)q_c + 64q_w \sin^2 \frac{\pi H_w}{2H} \sin^2 \frac{\pi H_w}{H}}} \tag{36}$$

2.3.2. Sufficient Condition

If a wellbore is destabilized, its total potential energy function equation satisfies the standard form of the cusp catastrophe model. To meet the condition

$$\Delta = 4m^3 + 27n^2 \leq 0 \tag{37}$$

$m$ must be negative. Combined with Equation (27), it can be seen that

$$A_2 \leq 0 \tag{38}$$

by substituting Equation (24) into Equation (38), we determine that the calculation result is the same as Equation (35), so Equation (35) is a sufficient condition for the occurrence of the vertical instability of the wellbore. The critical depth of wellbore destabilization is also the same as Equation (36).

### 2.3.3. Equation for Critical Depth of Wellbore Destabilization

According to Sections 2.3.1 and 2.3.2, the sufficient and necessary conditions for vertical instability to occur are that the depth of the wellbore is within the range shown in Equation (35), and Equation (36) is the critical depth for the occurrence of vertical instability of the wellbore.

## 3. Analysis and Discussion

### 3.1. Comparison Analyses

In this part, several representative drilling wellbore projects were selected, and the critical depth values before and after using optimization measures were calculated. The critical depth before adopting optimization measures was calculated based on reference [23]. The critical depth after adopting optimization measures was calculated by the Equation (36) in this article. The selected representative projects included the Panji west wind wellbore, the Keke gai coal mine wind wellbore, the Banji coal mine auxiliary wellbore, and the Banji coal mine main wellbore. The comparison calculation basis and process are shown in Tables 2 and 3. The calculation results are shown in Figure 4.

**Table 2.** Critical depth equations for different construction methods.

| | Traditional Methods | Optimized Methods |
|---|---|---|
| Restrictive conditions | With both ends hinged | Upper end hinged lower end fixed support |
| Sources of information | Literature [23] | Equation (36) |
| Critical depth formulas | $H_{cr1} = \sqrt[3]{\dfrac{2EI\pi^4}{q_c\pi^2 + 4q_w\sin^2\frac{\pi H_w}{H}}}$ | $H_{cr2} = \sqrt[3]{\dfrac{41EI\pi^4}{(10\pi^2-32)q_c + 64q_w\sin^2\frac{\pi H_w}{2H}\sin^2\frac{\pi H_w}{H}}}$ |

**Table 3.** Comparison calculation process of critical depth before and after optimization measures for representative engineering use.

| Shaft Walls | Panji West Wind Wellbore | Kekegai Coal Mine Wind Wellbore | Banji Coal Mine Auxiliary Wellbore | Banji Coal Mine Main Wellbore |
|---|---|---|---|---|
| Sources | Literature [30] | Literature [31] | Literature [32] | Literature [23] |
| $E$ (GPa) | 35.5 | 38.89 | 38.15 | 38.85 |
| $I$ (m$^4$) | 126.5 | 68.26 | 203.33 | 139.66 |
| $D$ (m) | 9 | 7.2 | 9.3 | 8.3 |
| $d$ (m) | 7 | 6 | 7.6 | 6.5 |
| $q_c$ (N/m) | 457,008 | 328,159 | 660,000 | 469,000 |
| $q_w$ (N/m) | 376,957 | 276,948 | 444,348 | 351,000 |
| $H_w$ (m) | 222.2 | 215.2 | 333.3 | 291.703 |
| Depths of wellbore $H$ (m) | 508 | 542.5 | 638.084 | 659.675 |
| Critical depths under the traditional method $H_{cr1}$ (m) | 527.28 | 495.98 | 566.95 | 563.58 |
| Stability under the traditional method | Stable | Unstable | Unstable | Unstable |
| Critical depths under the optimized method $H_{cr2}$ (m) | 766.11 | 729.06 | 805.52 | 817.55 |
| Stability under the optimized method | Stable | Stable | Stable | Stable |

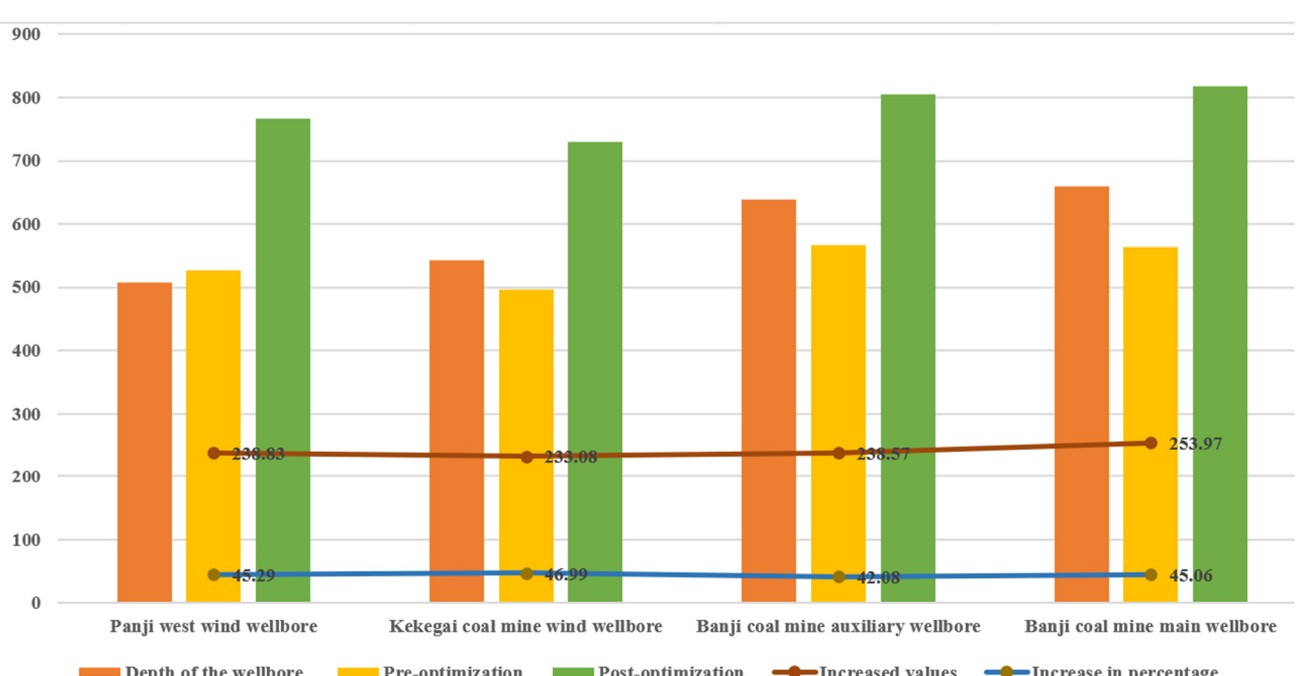

**Figure 4.** Analysis diagram of critical depth growth after optimization of each wellbore.

By comparing the calculation results of the critical depth of instability before and after the optimized construction process in Table 3, it can be seen that the wellbore stability changed significantly due to the different constraints under different construction processes. The critical depth of the wellbore instability calculated under the optimized construction technology was significantly higher than that under traditional construction technology.

In the theoretical analysis, the critical depths of instability of the Panji west wind wellbore, Keke gai coal mine wind wellbore, the Banji coal mine auxiliary wellbore, and the Banji coal mine main wellbore were previously increased by 45.29%, 46.99%, 42.08%, and 45.06%, respectively, after adopting the optimized measures. The differences are 3.9%, 5.6%, 0.69%, and 3.67%, respectively, compared with the results of the literature [33], which shows that the optimized construction measures are reasonable and reliable to improve the stability of wellbores by around 41.39 $\pm$ 5%.

### 3.2. Discussion on Critical Depth Influence Factors

From Equation (29), it can be seen that the factors affecting the critical depth of wellbores are elastic modulus $E$, the self-weight of the wellbore $q_c$, the thickness of the wellbore $\triangle$, and the height of counterweight water $H_w$. In this section, a two-by-two combination of single-factor analysis of the effects of the elastic modulus, the self-weight of the wellbore, the thickness of wellbore, and the height of counterweight water on the critical depth was performed by using the control variable method.

Taking the Banji coal mine auxiliary wellbore as an example, the depth of the wellbore is 638.084 m, the minimum height of the counterweight water is 333.3 m, the external diameter is 9.3 m, the internal diameter is 7.6 m, the elastic modulus is 38.15 Gpa, the moment of inertia of the cross-section is 202.65 m$^4$, the self-weight per unit length is $6.6 \times 10^5$ N/m, and the weight of counterweight water per unit length is $4.44 \times 10^5$ N/m. We use the above parameters as a baseline to distribute the change in the elastic modulus of the wellbore $E$ from 33.65 Gpa to 43.15 Gpa at intervals of 0.5 Gpa, the self-weight of the wellbore $q_c$ from 570,000 N/m to 760,000 N/m at intervals of 10,000 Gpa, the thickness of the wellbore $\triangle$ from 760 mm to 950 mm at intervals of 10 mm, and the height of counterweight water $H_w$ from 333.3 m to 618.3 m at intervals of 15 m to calculate and analyze the change of the critical instability depth of the deep wellbore structure after adopting the optimized construction measure. To briefly analyze the trend of any two influencing factors on the

critical depth of wellbore instability $H_{cr}$, the analytical diagram of influencing factors is arranged and drawn according to Table 4.

**Table 4.** Influence factor combination analysis matrix.

|  | $E$ | $q_c$ | $H_w$ | $\triangle$ |
|---|---|---|---|---|
| $E$ | $E$-$E$ |  |  |  |
| $q_c$ | $E$-$q_c$ | $q_c$-$q_c$ |  |  |
| $H_w$ | $E$-$H_w$ | $q_c$-$H_w$ | $H_w$-$H_w$ |  |
| $\triangle$ | $E$-$\triangle$ | $q_c$-$\triangle$ | $H_w$-$\triangle$ | $\triangle$-$\triangle$ |

As can be seen in Table 4, the combinations $E$-$E$, $q_c$-$q_c$, $H_w$-$H_w$, and $\triangle$-$\triangle$ in the table are considered duplicates because they are the same influencing factors. In the blank section above the duplicates, the combinations of any two influencing factors are duplicated with all those below the duplicates in this section. It is sufficient to consider analyzing only the following six factor combinations: $E$-$q_c$, $E$-$H_w$, $E$-$H_w$, $q_c$-$H_w$, $q_c$-$\triangle$, and $H_w$-$\triangle$. The calculation results are shown in Figure 5.

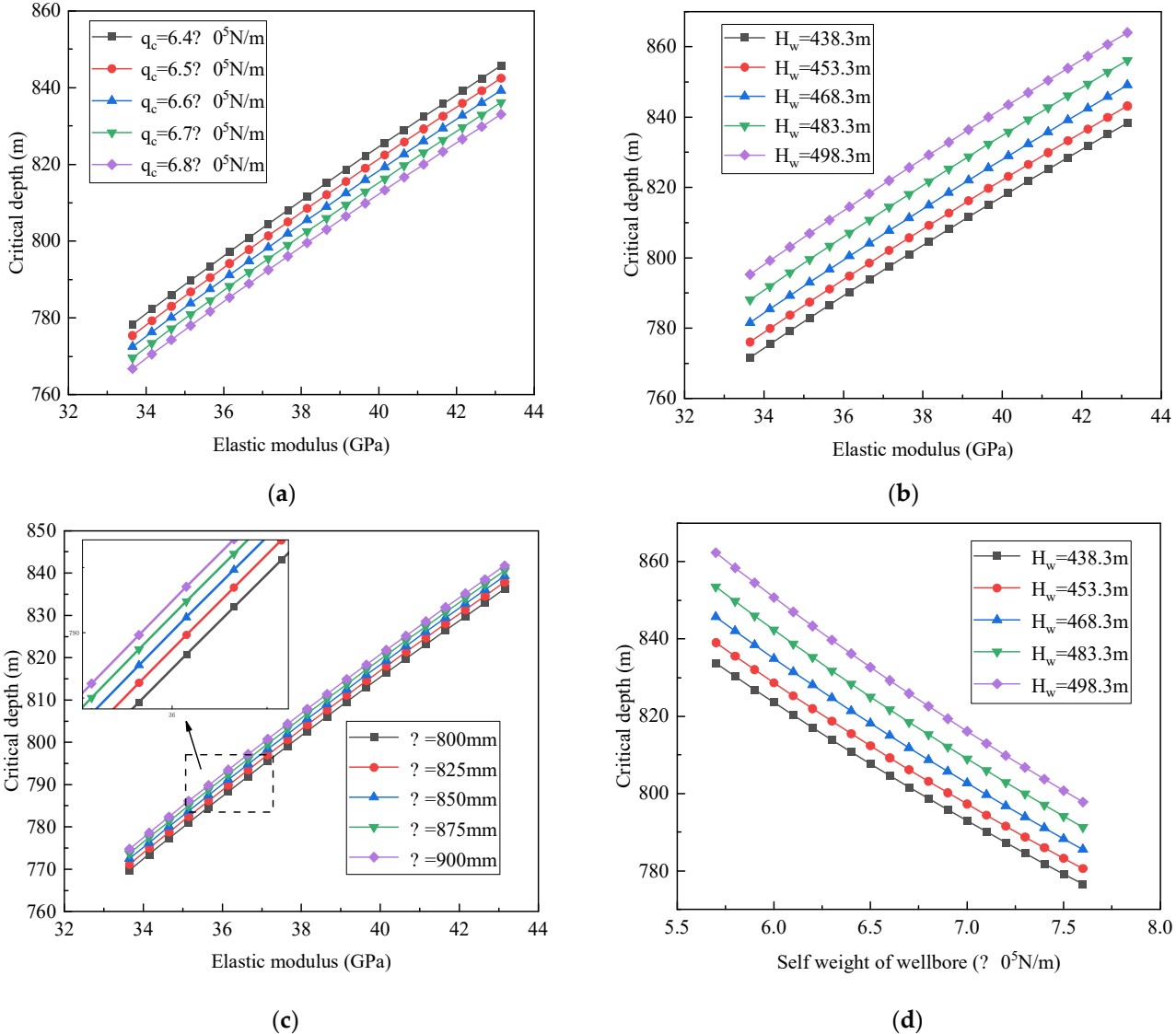

**Figure 5.** *Cont.*

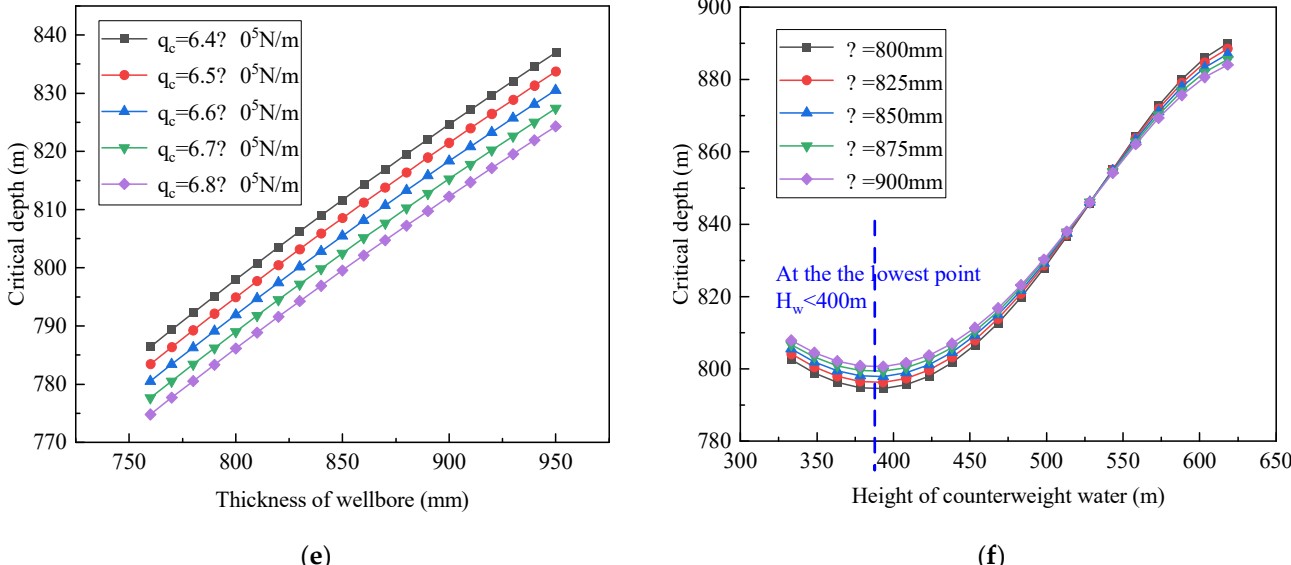

**Figure 5.** Effect of each factor on the critical depth of wellbore. (**a**) Effect of $E$ and $q_c$ on critical depths. (**b**) Effect of $E$ and $H_w$ on critical depths. (**c**) Effect of $E$ and $\triangle$ on critical depths. (**d**) Effect of $q_c$ and $H_w$ on critical depths. (**e**) Effect of $\triangle$ and $q_c$ on critical depths. (**f**) Effect of $H_w$ and $\triangle$ on critical depths.

From Figure 5a–c, it can be seen that the values of the critical depths of the wellbore increase with the increase in elastic modulus regardless of any self-weight of the wellbore, the height of counterweight water, and the thickness of the wellbore. Under a different elastic modulus, the critical depths of the wellbore instability decrease with the self-weight of wellbore and increase with the increase in the thickness of the wellbore, and the degree of increase tends to be slow. When the injection heights of counterweight water are 483.3 m, 453.3 m, 468.3 m, 483.3 m, and 498.3 m, the critical depths of the wellbore instability increase with the increase of counterweight water height, and the degree of increase tends to become urgent.

From Figure 5d,e, it can be visualized that under different heights of counterweight water, the effects of the self-weight of the wellbore on the critical depths are all the same, and the critical depths of the wellbore decrease with the increase in the self-weight of the wellbore. Under different self-weights of the wall, the effects of the thickness of the wellbore on the critical depths are also the same, and the critical depths of the wellbore increase with the growth in the thickness of the wellbore.

As can be seen from Figure 5f, the critical depth of wellbore instability first decreases and then increases with the increase in the height of counterweight water, starting from the minimum counterweight water of 333.3 m. When the height of counterweight water is between 375 m and 400 m, the critical depth reaches the lowest point. When the height of counterweight water is injected above the lowest point, the critical depth increases sharply at first and then slowly with the increase in the height of counterweight water.

### 3.3. Multi-Factor Sensitivity Discussion

The sensitivity coefficient represents the relative measure of the degree of influence of uncertainty factors on evaluation indicators [34,35]. The formula for calculating the sensitivity coefficient is $s_{AF} = \frac{(\Delta A/A)}{(\Delta F/F)}$, where $\Delta A/A$ is the corresponding rate of change of the evaluation indicator $A$, and $\Delta F/F$ is the rate of change of the uncertainty factor $F$. The larger the absolute value of the sensitivity coefficient, the more sensitive the evaluation indicator $A$ is to the uncertainty factor $F$. It can be seen from Figure 5 that the critical depth of the wellbore structural instability after the adoption of optimized construction measures is affected by several factors to varying degrees. The critical depth of instability of the

wellbore structure after the optimization measures was affected by some factors to different degrees. The sensitivity coefficients of critical depth to each factor are shown in Figure 6.

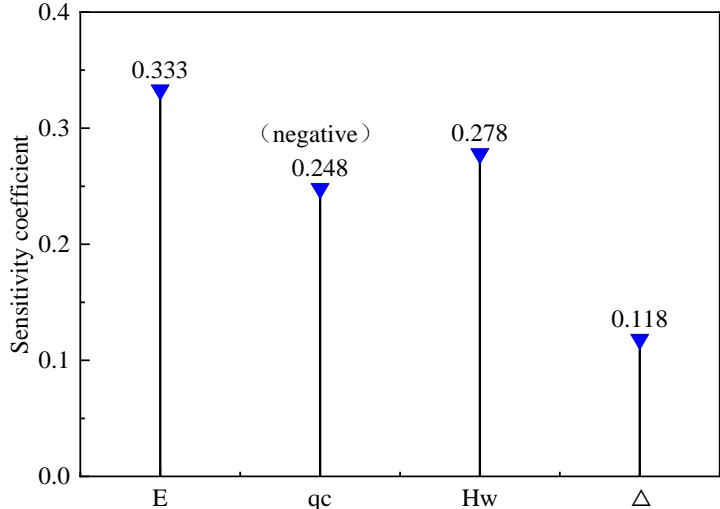

**Figure 6.** The sensitivity coefficient of the critical depth of wellbore to each influencing factor.

Figure 6 shows that in descending order, the effect degree of each factor on the critical depth of the wellbore instability is the elastic modulus (0.333) > the height of counterweight water (0.278) > the self-weight of the wellbore (0.248) > the thickness of the wellbore (0.118). Additionally, the sensitivity coefficients of critical depth of the wellbore instability to the elastic modulus, height of counterweight water, and thickness of the wellbore are all positive, while the sensitivity coefficient of the self-weight of the wellbore is negative, indicating that the critical depth of the wellbore instability changes in the same direction as the elastic modulus, height of counterweight water, and thickness of the wellbore and changes in the opposite direction of the self-weight of the wellbore.

## 4. Numerical Calculation Verification

### 4.1. Principles of Stability Numerical Calculation and Numerical Models

Euler stability was determined to be the cause of the wellbore's stability issue. The generalized eigenvalue solution of Euler stability in numerical analysis and the incremental solution of extreme stability were used to determine this stability. Literature [36] provides a detailed explanation of the numerical calculation process for the stability of wellbore structures. Based on numerical calculation principles, the finite element software ABAQUS can be used to calculate numerical simulation eigenvalue and analyze the structural stability. When the eigenvalue $\lambda$ satisfies $\lambda < 0$ or $\lambda > 1$, the wellbore is vertically stable, and when $0 < \lambda < 1$, the wellbore will be unstable.

The numerical simulation section of this paper is based on the Banji coal mine auxiliary wellbore as the engineering basis and establishes a physical model of the second stage of the wellbore construction process, which is the stage where the wellbore floated to the bottom but is not filled and fixed. Since a wellbore is cylindrical in shape, with constant inner and outer diameters, it can be regarded as an axisymmetric model. The establishment of the entire wellbore structure model is done through the input file. According to the characteristics of the wellbore structure, the entire wellbore is divided into two layers: the lower layer is the bottom of wellbore, and the upper layer is the joint of wellbore. The establishment process of the wellbore is shown below.

(1)  Definition of nodes and node sets. The node types are defined by *Node and Nset. A total of 15,433 nodes are defined in the entire wellbore model, and the height of wellbore model is 638.084 m. The nodes at the bottom of the wellbore are named "BOTTOM", and the nodes at the top of the wellbore are named "TOP".

(2) Definition of elements and elements sets. In this structure, the elements in the bottommost of the wellbore are defined using three-node triangular linear film strain linear shell elements, referred to as S3R. All other elements are specified to use four-node quadrilateral linear reduced integration shell elements, referred to as S4R.

(3) Define surfaces. Surfaces are defined by the *Surface command, which describes the faces composed of elements. For all elements, their inner surfaces are named "In", and their outer surfaces are named Out.

(4) Define materials. Materials are defined by the Materials command. The wellbore structure is made of reinforced concrete. The concrete is graded C70. The elastic modulus of concrete is 38.15 GPa, and its Poisson's ratio is 0.285. The reinforcing steel has an elastic modulus of 210 GPa, and its Poisson's ratio is also 0.285.

(5) Define shell parameters. Defined by the Shell section command, the thickness of the shell is 0.85 m.

(6) Definition of loads. The height of counterweight water is 333.3 m, and the height of mud is 638.084 m.

(7) Define boundaries. The *Boundary command is used to define boundary conditions. We define the corresponding constraints on BOTTOM and TOP.

(8) Define buckling. Buckling is defined by the Buckle Step command. We set the number of eigenvalues requested to five.

The finite element calculation model of the equal cross-sectional wellbore, established according to above definitions, is shown in Figure 7a. The types of elements used in the finite element model are depicted in Figure 7b,c.

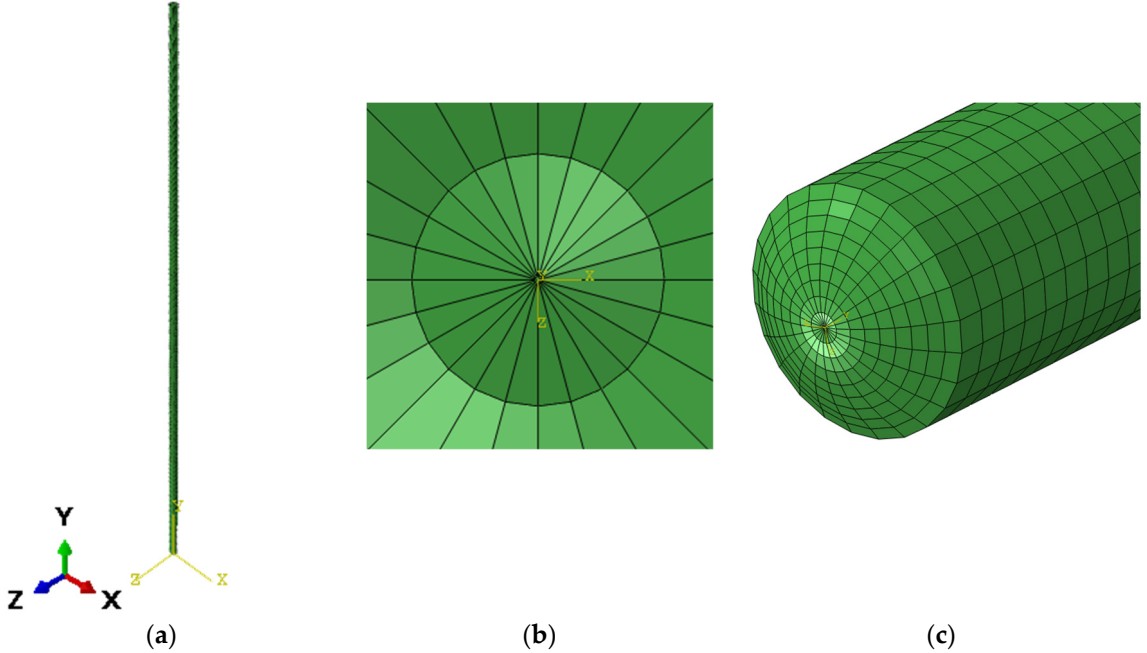

**Figure 7.** Finite element model of the entire wellbore and the types of elements used. (**a**) Numerical model of the complete wellbore. (**b**) Element type of S3R. (**c**) Element type of S4R.

*4.2. Comparative Analysis of Numerical Models*

The stabilities of the wellbore structure before and after the optimization of construction measures are compared by numerical calculation techniques in ABAQUS 2020. The two finite element models are consistent in the definition of material parameters, section characteristics, analysis steps and load structures, but the constraint conditions at the bottom of the wellbore have changed. The constraint comparison of the wellbore models before and after optimization is shown in Table 5.

**Table 5.** Comparison of constraints of wellbore models before and after optimization.

| Construction Techniques | Constraints on BOTTOM | Constraints on TOP |
|---|---|---|
| Traditional construction technology | U1, U2, U3 | U1, U3, UR2 |
| Optimized construction technology | U1, U2, U3, UR1, UR2, UR3 | U1, U3, UR2 |

The boundary conditions of the finite element model of the drilling wellbore established under the two constraints in Table 5 are shown in Figure 8.

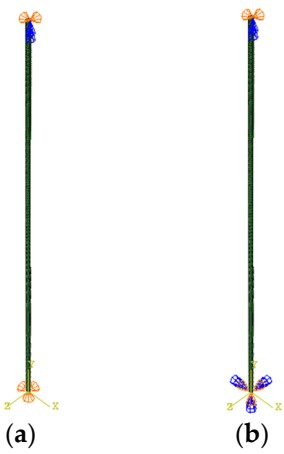

(**a**)        (**b**)

**Figure 8.** Comparative boundary condition analysis of wellbore finite element models before and after optimization. (**a**) Boundary condition of wellbore before optimization. (**b**) Boundary condition of wellbore after optimization.

The cloud diagram obtained after the model was submitted for analysis is shown in Figure 9.

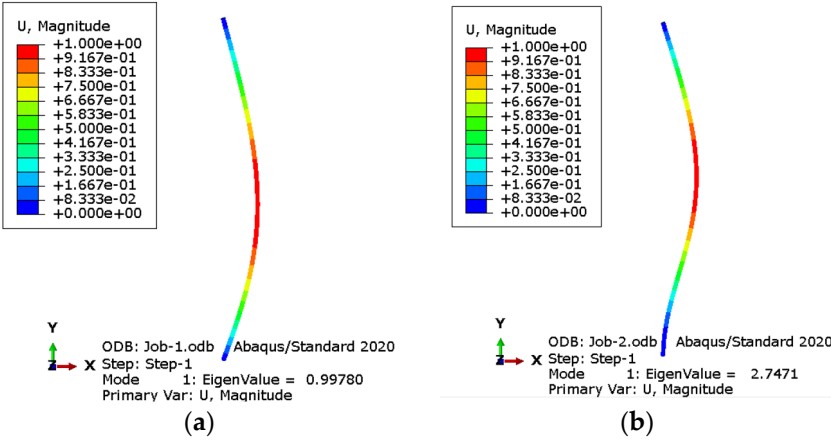

(**a**)        (**b**)

**Figure 9.** The displacement cloud diagram of the finite element model for wellbore structure. (**a**) Under conventional construction method. (**b**) Under optimized construction method.

As can be seen from Figure 9, the stress conditions of wellbore structures before and after optimization are similar. The maximum displacement values for both cases are 1 m. However, due to the different locations of maximum deformation, the eigenvalue of the wellbore structure under the traditional method is 0.9978, which is less than 1, indicating that the wellbore structure is unstable. In contrast, the eigenvalue of the optimized wellbore structure is 2.7471, which is more than 1, indicating that the structure is stable. This is in agreement with the theoretical calculations of the wellbore structure's stability. Therefore, the optimization measure can significantly enhance the stability of wellbore structures.

**5. Conclusions**

(1) In this paper, it is proposed that cement mortar can fill the the bottom of the well in advance to improve the stability of the wellbore before the wall is suspended and sunk to the bottom. According to the contact characteristics of the bottom of the wall after optimization, the wall cylinder is regarded as an equal-section compression rod hinged at the upper end and solidly supported at the lower end, and based on the mechanism of cusp catastrophe characteristics, the critical depth of instability of an equal-section drilling wall in a non-full-water state is deduced from catastrophe theory.

(2) Compared with the traditional method, the critical depth of the wellbore structure after the optimization method can be increased by $41.39 \pm 5\%$. Through numerical simulation using a sub-well as an engineering example, the eigenvalue of the traditional method is 0.9978, which is smaller than 1, indicating that the wall is unstable, while the eigenvalue of the optimization method is 2.7471, which is larger than 1, indicating that the wall is stable, which is in line with the theoretical calculation. Both the theoretical and numerical analysis reveal that the optimization method is reliable and applicable.

(3) The critical depth of the wellbore structure by using optimization measures is influenced by multiple factors. It increases with the increase in the elastic modulus and the thickness of the wellbore, decreases with the increase in the self-weight of the wellbore, and shows a trend of first decreasing and then increasing with the increase in the height of counterweight water.

(4) The critical depth of the wellbore structure by using optimization measures is influenced by multiple factors to varying degrees. The degree to which they are affected in descending order is elastic modulus, counterweight water height, self-weight of the wellbore, and thickness of the wellbore.

(5) In practical engineering, it is of great practical significance to use high-strength steel with a higher modulus of elasticity or increase the height of appropriate counterweight water and reduce the thickness of internal and external steel plates to improve the stability of the wellbore.

**Author Contributions:** Conceptualization, R.P.; methodology, J.L. and H.C.; software, J.L.; validation, R.P.; formal analysis, R.P.; investigation, H.F.; resources, R.P.; data curation, H.F.; writing—original draft preparation, R.P.; writing—review and editing, J.L.; visualization, J.L.; supervision, J.L. and H.C.; project administration, J.L. and H.C. All authors have read and agreed to the published version of the manuscript.

**Funding:** This research was funded by Anhui Province Key Laboratory of Building Structure and Under-ground Engineering (Hefei), Anhui Jianzhu University (KLBSUE-2022-01).

**Institutional Review Board Statement:** Not applicable.

**Informed Consent Statement:** Not applicable.

**Data Availability Statement:** The data presented in this study are available on request from the corresponding author (privacy).

**Conflicts of Interest:** The authors declare no conflicts of interest.

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
