# Peer review of "Study on the Vertical Stability of Drilling Wellbore under Optimized Constraints"

_applsci, doi:10.3390/app14062317_

Round 1

Reviewer 1 Report

Comments and Suggestions for Authors

This is the first review of the manuscript: Study on the vertical stability of drilling wellbore under optimized constraints.

The subject is interesting, although not fully new.

The manuscript is slightly ambiguous in some parts. The main problem throughout the manuscript is that the authors are using the word “wellbore” for the actual hole and for the machine that is making the hole, thus causing a lot of confusion. Even when it is distinguishable, still the way of describing the process is unclear/difficult to understand.

Therefore, authors should make an effort and revise the manuscript.

Comments on the Quality of English Language

The main problem throughout the manuscript is that the authors are using the word “wellbore” for the actual hole and for the machine that is making the hole, thus causing a lot of confusion. Even when it is distinguishable, still the way of describing the process is unclear/difficult to understand.

Author Response

Thank you very much for taking the time to review this document. I have submitted the responses in the attached file below. The replies and modifications are all highlighted in blue. Please check the detailed responses and the corresponding changes highlighted in the revised submission.

Reviewer 2 Report

Comments and Suggestions for Authors

The manuscript “Study on the vertical stability of drilling wellbore under optimized constraints” corresponds to the topic of stability of drilling wellbore structure. The authors presented research with sufficient mathematical justification of the wellbore optimization method.

However, some aspects can be improved:

1.       Please use more up-to-date literature sources. Currently, 48% (15 out of 31) are older than 5 years. Please update the reference list with more recent articles.

2.       Can you please explain in the manuscript text what variables x, m, and n mean? (Page 4, equation 3,4). What is their physical nature?

3.       Considering the large number of various variables and symbols used in Chapter 2 and especially Subchapter 2.2, it is highly recommended to form the list of symbols at the beginning of the manuscript with an explanation of what each symbol means.

4.       Authors mentioned that “E” is elastic modulus. But they didn’t specify the elastic modulus of what material. Probably some geological rock is considered. On page 9, the authors mentioned that “the elastic modulus of wellbore E…”. There are doubts that wellbore has some elastic modulus because it is a hole in geological rock. Please reconsider the sentences with elastic modulus and specify the material elastic modulus characterize.

5.       Chapter 4. In this chapter, there is very little information about the set-up of FEM calculations. It is highly recommended to add a figure with a detailed set-up of calculation and a table of boundary conditions. Also, it needs to be made clear how the results of optimizations were considered in the model set-up, and what dynamic mathematical models, element types and other so on. The information about FEM is not sufficient for the replication of these results.

Author Response

(The authors gave the same response as above.)

Reviewer 3 Report

Comments and Suggestions for Authors

After a detailed study of the article, it can be concluded that its topic is interesting both from the point of view of theory and the practical use of the knowledge gained. In terms of substantive content, I have no major comments, only in chapter 4. the reader would undoubtedly welcome a more detailed description of the FEM model in the ABAQUS program, including the type of finite element used.

Regarding the formal organization of the text, I recommend considering the following:

Most of the listed mathematical relationships are numbered, some are listed only as part of the text, e.g. on page 4. It would perhaps be more convenient for the reader if these equations were also numbered, and if they cannot be considered generally known, then the relevant references should also be given.

In cases where numbered equations are part of a sentence, it is not appropriate to start the text after the equation with a capital letter, as is the case, for example, with equation (1) on pages 2 and 3.

References to the literature, e.g. in the introduction, use a hyphen, not a dash, in curly brackets, in accordance with the template.

Author Response

(The authors gave the same response as above.)
